# A Tale of Two Sophias: A Proposal for Critical Posthuman Youth Work, and Why We Need It

**Maria Pisani**

Department of Youth & Community Studies, Faculty for Social Wellbeing, University of Malta, MSD2080 Msida, Malta; maria.pisani@um.edu.mt

**Abstract:** This paper begins by recounting a tale of two Sophias: a humanoid robot and an 'illegal' baby immigrant. The tale of two Sophias locates my initial ideas for reflecting on how critical posthumanism might contribute to youth work theory and practice. In this paper I position youth work as a philosophical encounter, whilst also questioning the humanist legacy that lies at the heart of youth work theory. Drawing on the work of Rosi Braidotti and other critical posthuman feminists, I consider how youth work might respond to the posthuman predicament marked by the intersecting forces of advanced capitalism and growing inequalities, the fourth industrial revolution, the digital divide, and advances in Artificial Intelligence, climate change, and environmental destruction. I conclude by providing some reflections on how critical posthuman theory may provide a lens through which young people might consider what it means to be human in the technologically mediated Anthropocene, and also as a paradigm for embracing new possibilities and a praxis of hope.

**Keywords:** youth work; critical posthuman theory; Anthropocene; challenges; climate change

## 1. Introduction

This paper documents a tale of two Sophias: one a humanoid robot, the other an 'illegal' baby immigrant. It has taken me a while to write this paper. When I first formulated the title in my head some years ago, my world, and I would hazard a guess, your world, was a little different. COVID-19 not only demonstrated our shared vulnerability, but also exposed the deep structural violence that has ensured that 'our' vulnerability is not shared equally. The global pandemic brought with it loss of life, suffering, disruption, and stark warnings from the scientific community. The human species, and the hegemonic economic paradigm that has been advanced at any cost, is not only responsible for a global health crisis, but also for the climate and biodiversity crisis that challenges the survival of the human species, and perhaps also the planet that we call home [1]. Indeed, whilst young people have been at the forefront of climate activism, youth work has been 'reticent' in its response [2,3].

The future of globalization, the global economy, in/equality and economic growth, rapid technological advances, war, migration and mobility, climate change, borders and the nation state, increased interdependence and retreat—these contemporary challenges are entangled and evolving. Within the European context, such issues are recognized as key challenges for youth work in the 21st century [2,4].

In light of these challenges, and the tale of two Sophias that I am still to recount, this paper seeks to explore how critical posthuman theory might provide a lens through which youth work might explore what it means to be human in the 21st century, to understand and respond to the evolving questions, realities, and challenges that 'we' face today, and as an opportunity to imagine different possibilities, choices, and hope.

In their respected work on philosophy in youth and community work, Seal and Frost [5] claim that philosophy inspires what we do as youth workers. The authors demonstrate the value in approaching youth work practice philosophically, to embrace considered

thinking, and to explore how philosophy might pose questions, and challenge the common-sense assumptions that ground our work and practice.

To date, my research and my practice has been grounded in a critical approach, drawing on critical race theory, feminism and intersectionalities [6,7], postcolonial studies [8,9], and critical pedagogy [9–11]. Always, my work has sought to embrace the borderlands [12] and has taken an unapologetically political approach, committed to social justice. These familiar resources that continue to serve as an anchor for my thinking and practice, though, have not evolved in a way that captures and provides an adequate framework for under-standing and navigating the contemporary challenges touched on above.

Perhaps never before has it been more right, or more urgent, to ask new questions, and some of these questions served as the premise of this paper: What does it mean to be human on a planet under the effects of capitalism on steroids, including a devastating climate crisis and the Sixth Extinction [13], economic and social divides, and advanced (sometimes exciting and lifesaving, sometimes frightening) technological developments? How are we to understand our relationship with other humans and non-humans with whom we share this planet, the broader 'community of practice' so to speak? [2]. These are just some of the vital questions that young people will need to reflect on as they imagine and shape the world that they want to live in. How might youth work—in our thinking and in our action—respond to the new problems and possibilities arising from what Braidotti [14] describes as the posthuman predicament?

In this paper I consider the critical posthuman paradigm as a theoretical tool to (I hope) inspire a new way of thinking about youth work practice, as a lens with which to critically explore our own journey of becoming as we collectively and differentially navigate the posthuman predicament. The paper seeks to explore how critical posthuman theory might contribute to an ethically and politically committed youth work practice that might imagine and work towards new possibilities for human and non-human co-existence.

To do this, I provide a brief overview of the key theoretical thoughts that inspired this paper, primarily drawing on the work of critical posthuman feminist theorists, in particular, Rosi Braidotti [14–16]. This is followed by a tale of two Sophias: baby Sophia, an 'illegal' baby, born on the Mediterranean Sea, and Sophia the humanoid cyborg. The tale of two Sophias frames my ideas moving forward. Drawing on youth work literature, with a focus on youth work as a philosophical encounter, I then present a sketch, or outline, of how I think critical posthuman theory might lend itself to a critical, affirmative, and transformative ethical framework for youth work practice that may respond to the posthuman predicament.

## 2. Philosophia, the Love of Wisdom

Cooper [17], Williamson [18], and others have demonstrated the importance of theory in articulating the point of convergence, and making sense of the diversity of youth work practice and contexts. The tale of two Sophias—a story I am yet to recount—served as the beginning of my own questioning: Can the theories and philosophical paradigms that have informed my own understanding and ethos of youth work and youth work praxis respond to the challenges and realities of the present and the future? Do these frameworks ask all of the relevant and urgent questions? In the summer of 2019, I visited Kakuma refugee camp in Kenya. I recall walking into a room to meet a group of young people, who were watching Money Heist on Netflix. It struck me because at the time many of the young people I knew in Malta were also watching the show—I reflected on the extent and apparent frictionless reach and influence of global media and global youth culture. Images and ideas are flashed across screens, transcending borders with ease. Paradoxically, the young people I met in Kakuma had never left the camp they called home, because they were not allowed to. That night I was scrolling through my own smart phone, when the President of France had tweeted 'The Amazon rainforest—the lungs which produce 20 percent of our planet's oxygen—is on fire. . . '. Macron was angry with Bolsonaro, accusing him of not doing enough to tackle global warming [19]. As I lay on my bed within the confines of Kakuma

camp, the contradictions of geographical and sovereign boundaries—and their effects and affects—could not have appeared more apparent, toxic, entangled, or urgent. It was also clear to me that the theories to which I have turned as a guide, a source of inspiration, and also familiar comfort, I felt, were not only lacking, but strangely out of touch with the realities that seemed to me to be historically recognizable, and yet very particular, and very contemporary. What follows is not a negation of those ideas and theoretical paradigms that have framed my thinking and my practice, but rather, a revisiting, through a critical posthuman lens.

I started from the position that the theoretical paradigms on which I have depended to make sense of the world and inform my theorization around youth work are falling short in explaining and addressing the painful challenges we face at this time. These challenges are part of living in a globalized world under advanced capitalism and include (but are not limited to) the climate crisis and ecological collapse, the present and future of Artificial Intelligence, the so called 'refugee crisis' and racism, a growing gap between the haves and the have nots, and indeed, the global COVID-19 pandemic. Importantly, these issues are explicitly connected to each other. Others [20] have also highlighted the importance of looking at the macro context and how it shapes youth work and its practice, urging youth workers to rethink their methods, analysis, and practices to address the 'social, cultural and economic' effects of global capitalism on the lives of young people around the world in such a way that is 'responsive to 21st Century conditions' [20]. Advancing a model of Global Youth Work (GYW), Sallah, Ogunnusi, and Kennedy [21] call for an approach that is 'concerned with how the concept and process of globalization impacts on young people's realities' [21]. Sallah and Cooper [22] draw attention to globalization as 'the world coming together due to closer economic, cultural, environmental, political and technological interactions resulting in global interdependence', and position GYW as the 'process of working with young people to make the personal, local, national and global connections and if need be, take action...'. Echoing this position, Belton argues that Global Youth Work is geared towards the betterment of the young person and 'wider global society' [23].

My intention is not to detract from this line of thought, but to build on it. It would seem to me that an element of human exceptionalism lies at the heart of youth work, a Godly power afforded to Anthropos that also lies at the heart of ecological destruction, risking not only the life of humankind, but the life of the whole planet. It dawns on me today that there are limitations to the notion that we might understand the lives of young people, and their relationships with and in the world, by limiting our theories to the study of society and social relationships. It is such hegemonic anthropocentric ideas that have fed into the unquestioned belief that the planet is simply a resource to be used and abused by 'Man'. To be clear, 'Man' here does not refer to all of humanity; baby Sophia, and billions of others with whom she shares this planet, were never really considered fully human in the first place. And so, I would argue that understanding the effects—and the affects—of the human-made ecological and environmental crisis requires much more than an add and stir approach to youth work theory and praxis. We need new theories, a new epistemological lens and ethics of care that will respond to the devastating ecological, social, and economic effects (which include the extinction of our species) of advanced capitalism. Drawing on the work of Latour, I have found it useful to think about and understand this planet that we call home as a 'totality of living beings and materials that were made together, that cannot live apart, and from which humans can't extract themselves' [24].

Located within advanced capitalism, Braidotti [14] understands the 'posthuman' as resting on a critique of two aspects of the Humanist ideal of 'Man'. The first is that which places the Western male white able-bodied subject as the ideal subject. The second adopts a postanthroprocentric lens which questions the notion that 'man' lies at the center of the universe—that, as a species, humans are exceptional, and that all other beings (animal and plant) are simply a means to an end. Critical posthuman theory lies at this intersection.

Within the posthuman critique, Vitruvian man, historically held at the center of mankind as the measure of all things human, and central to the binary logic of the 'other' as the gendered/racialized/sexualized/naturalized subordinate, is deposed, untethered from his universalistic stance and held to account. But this does not mean that this paradigm should be thrown out with the proverbial baby. Humanism has indeed been the driver of a 'civilizing project', one that has elevated some humans above others, and positioned the less-than-Human-human-others, animals, and the environment as an exploitable source, a means to an end. And humanism has also advanced the ideals of freedom, independence, social justice, equality, and human rights, and these values remain important, particularly for humans like baby Sophia. These values also lie at the heart of youth work theory and practice [23]. In my own activism, I have experienced how human rights can be an effective tool in holding power to account, particularly for those groups historically negated as human in the first place, including, but not limited to, the young refugees with whom I work. But this does not mean that we should ignore the tensions around human rights, notions of the universal and a singular human identity, and how the idea structures and bounds our understanding of global politics and possibilities for change (see, for example, Baxi [25]). Likewise, and as I have demonstrated elsewhere, the 'right to rights' should not be taken for granted, and this includes within youth work theorization (see, for example, Pisani [26,27]). The tale of baby Sophia demonstrates how the basic human rights of illegalized racialized bodies are denied every day. Sophia's rights, her very being, along with those of millions of others, continue to be violated and have never been placed at the center. In decentering Man, the dualism that has justified the colonization of land, resources, naturalized bodies including indigenous people, and sexualized bodies—those cast as nature rather than human—is also dislodged, emancipating the naturalized other. Such theorization reaffirms my own grounding in, and reliance on, anti-racist, postcolonial, feminist, indigenous, and other marginalized knowledge in thinking through the youth work praxis (see also Skott-Myhre [28]; Pisani [29] Sallah [30]).

Braidotti [16] locates the decentering of the human species, and the posthuman predicament within the broader context of the Anthropocene. A term originally coined by Nobel prize winner Paul Crutzen, the term Anthropocene refers to the epoch of large-scale, human-driven changes to the Earth and its ecosystem. Evidence for the Anthropocene is well documented, but also evident for all of us to see, in pollution, physical changes as a result of urbanization, sea level rise, global warming, and biological changes linked to species invasion and extinction—the list goes on [31]. Globalization, climate change, and the destruction of the biosphere has facilitated the spread of new pathogens, as they cross species and borders, including, but not limited to, COVID-19.

One might argue that in the destruction of nature, we destroy ourselves, ergo, if we are to survive as a species, then concerns for the environment and the non-human require ethical and political consideration. But it seems to me that such an approach echoes the familiar anthropocentric lens that assumes human exceptionalism whilst simultaneously dehumanizing populations along racialized, gendered, heteronormative, geographical, and economic axes of power. Critical posthuman theory advances a 'radical repositioning on the part of the subject', that requires taking a step away from the hierarchal relations that continues to privilege some 'men' more than others, whilst *simultaneously* questioning the notion of human exceptionalism. It is this approach, I am arguing here, that would provide youth work theory with the possibility to engage with issues such as globalization, inequalities, capitalism, and climate change in all their complexity.

Whilst acknowledging the concerns of liberal and humanist thinkers (she cites Habermas and Fukuyama among others) who have expressed a panic not unlike that which I felt on learning of the rights to be afforded to Sophia the robot, Braidotti [15] invites the reader to reflect on the benefits of the posthuman turn. Under advanced capitalism, all 'life'—including the Vitruvian man—has been thrown off-center, commodified and subsumed into global networks of control and challenged by scientific advances and practices that are

already 'deeply inhumane' [15]. Braidotti positions the posthuman dimension of postanthropocentism as a move that deconstructs species supremacy, whilst also challenging the

> 'notion of human nature, *anthropos* and *bio*, as categorically distinct from the life of animals and non-humans, or *zoe*. What comes to the fore instead is a nature-culture continuum in the very embodied structure of the extended self'. [15] (italics in original text)

The posthuman convergence exposes the limitations inherent to the nature/culture binary, and highlights our interconnections with both human and non-human others, offering new epistemological ways of thinking. Rather than equating subjectivity with the rational consciousness of the individual subject, Braidotti [14] draws on feminist theory and Deleuze and Guattaris' neo-materialist philosophy to advance a vitalist materialist process-oriented political ontology 'based on immanence and becoming, defined as a creative praxis of actualization of the virtual [a subjectivity which] is structured by ontological relationality [with] the power to affect and be affected' (p. 62).

This is a radical move, an approach that I admit to finding challenging, not least because 'we' are asked to let go of the privileges and power that some of us have learnt to take for granted as a species, as humans. The tale of baby Sophia, however, will remind us that 'we' have not all benefited in the same way from these privileges. Importantly, Braidotti's posthuman approach should not be understood as inhumanism, neither a relativist approach nor one that constitutes a 'flat' ontology, but, rather, one that recognizes multiple situated perspectives, reflecting a politics of location, connecting *all* humans to non-human forces. Such an approach simultaneously moves away from human exceptionalism, whilst also recognizing and empowering the particularity and value of being human—of every human—at a time when 'Man' is also being dislodged from the center by advanced capitalism and the commodification of all living matter [15].

As a methodology, new materialism embraces the materiality of our shared planet, including the non-human actors, and adopts a politics and practice of change that responds to the challenges of the Anthropocene. Braidotti and other leading scholars (including Donna Haraway, Elizabeth Grosz, and Karen Barad, among others) seek to return to 'matter'. In doing so, they draw upon an eclectic range of thought that transcends disciplinary boundaries across the human and national sciences, including (but not limited to) feminist theories, queer theory, environmental studies, and critical race theory. This polycentric approach enables two important shifts: the first involves a theoretical turn away from the dualism that is so enduring in the humanist tradition, including nature-culture, whilst the second involves a shift away from the anthropocentric lens. Both of these have also informed much of youth work theory, and which, I will argue, represent limitations in our theorization in light of the contemporary challenges I document above. Questioning the stability of the individual, liberal subject, new materialist researchers call for a critical materialist focus on the planetary condition under late capitalism. Building on (but not dismissing) the work of social constructionism (including, for example, the construction of race, gender, and their intersections), this methodological approach emphasizes the complex and dynamic make-up of material and discursive productions and experiences of reality. In highlighting the limitations of the linguistic turn and the emphasis on language, culture and representation, new materialists look to explore material realities entangled with discursive practices. The emphasis on embodied circumstance looks to how material bodies (human and non-human, organisms and technologies), spaces and conditions interact and influence subject formation.

A new materialist ontology would understand young people as material individuals, with biological, evolving needs, living in a complex world shared with the animate and inanimate. Fox and Alldred [32] have adopted a new materialist approach to explore the affectivity of young bodies, and the flows that produce power and resistance, affecting what young bodies can and cannot do. All matter is understood as agentic and relational, with the potential to affect and be affected, influenced by, for example, globalized economic structures and cultural forces:

'... affect is not emotion and is not localised in individually separable bodies; affect is multiple, it is a force, a vitalist power, a capacity to act and to be acted upon; affect traverses all bodies and binds them together via cross-cutting flows'.

(Bozalek, Braidotti, Shefer, and Zembylas, [33]: p. 88)

Affect, then, is the ability to affect and to be affected, that extends beyond the human, as a transformative process of 'becoming'. New materialisms look to the relational aspects of politics, nature, human biology, and economics, and how they are enmeshed. The recognition that matter is co-existing, co-evolving with technology and culture, presents different possibilities for emancipation, political work, and transformation.

### 3. A Tale of Two Sophias: Sophia the Humanoid Robot

In November of 2018, Sophia, a humanoid robot, was the guest of honor at a conference in Malta. Sophia has been designed as aesthetically white, female, and youthful. During the conference, the Junior Minister for Social Innovation unveiled plans for a pilot project looking at developing a test that would ascertain whether Sofia and other robots with Artificial Intelligence (AI) are able to understand their legal rights and responsibilities as citizens. The pilot project, undertaken by the Government of Malta and the Artificial Intelligence (AI) company SingularityNET, announced the project at the Malta Blockchain Summit, a gathering of more than 5000 innovators and investors working in the field of digital technologies.

The head of the AI task force in Malta stated that granting citizenship rights to a robot in Malta would also grant rights and access to the European Union [34]. Sophia has already been granted citizenship by Saudi Arabia. The SingularityNET CEO was reported as being impressed by Malta's interest in citizenship for AI robots, adding that 'If a robot can visibly demonstrate a knowledge of its rights and responsibilities of being a citizen, then there us [sic] no reason it shouldn't be one.' [35]. The Junior Minister confirmed that the Government of Malta was applying 'certain citizenship conditions' to robots, adding that there was so much more to technology than AI, and that 'machine learning algorithms could be employed to solve complex problems that were previously beyond the reach of humans' [35].

'The Singularity' refers to the theoretical condition of the near future when machine intelligence will transcend human intelligence—radically transforming human lives and our very existence [36]. The development of 'superintelligence', expected to be greater than human minds in every quantifiable way, brings with it visions of the future that may lead to both apocalyptic change and technological salvation. Certainly, in different ways, Artificial Intelligence is already transforming our lives and raising new and urgent ethical questions: How might developments in AI affect human relationships? What would it mean for politics, for democracy? For labor and employment? And what about inclusion, issues around in/security? What will be the impacts on the planet? Who benefits? Who doesn't? Webb [37] argues that we are already treading a dangerous path, as the title of her book suggests, towards 'warping humanity'. Global Tech giants (Google, Microsoft, Amazon, Facebook, IBM, and Apple, the US 'G-MAFIA', and Chinese Baidu, Alibaba, and Tencent, collectively referred to as BAT) presently control the future of AI, and with it, the future of democracy, freedom of speech, and many other freedoms that some of us presently enjoy but take for granted. Whether driven and constrained by the short-term demands of capitalism (data are the new oil, look at how our use of social media provides data on consumption patterns), or the mining and refining of huge amounts of data (Cambridge Analytica and the manipulation of voters and election campaigns), the presence of AI in our lives today raises huge ethical questions—for philosophers, scientists, and also youth workers and young people. Albeit delayed, these questions are now being asked within the youth work context, looking at how AI impacts the lives of young people and how policy and practice should respond (see, for example, the CoE seminar report on Artificial Intelligence and its Impact on Young People [38]).

The fourth industrial revolution (4IR) characterized by the merging of the digital, biological, and physical worlds and developments in AI, brings with it a new epoch of economic disruption and uncertainty, including in relation to youth labor markets. Robots, automation, and digitalization will increasingly create job displacement and will impact, both negatively and in positive terms, the future of the human labor force. Although impossible to forecast, the ILO predicts that these impacts will not be shared equally, and whilst those young people who grew up as digital natives should be able to negotiate the shifting demands, the future does not look bright for low-skilled youth [39]. Of course, this should come as no surprise; there is no reason to believe that the 4IR should be able to redress existing growing global inequalities within the present capitalist hegemonic system, which is why such technological advancement brings with it renewed political and social challenges, not least for youth policy and youth work.

The Conclusions of the Council and of the Representatives of the Governments of the Member States meeting within the Council on Digital Youth Work [3] stress the need to bridge the digital divide (that is also mediated by gender, age, education, income, social groups, or geographic location, and their intersections), calling for equal opportunities for young people to enhance digital competences in order to participate in the 'information society' and 'enhance employability'. The report also highlights the role of youth work in engaging young people at risk of 'being left behind in a digitalised society', to empower young people to 'take responsibility and control of their digital identity' and 'address the challenges of convergence between the digital and physical environments' and the critical and responsible engagement with digital technologies for learning and participation in society. This convergence demonstrates the increasing fluidity between the human and technology (something that many of us will be able to identify with), and as noted in the CoE report, 'many young people perceive technology as another body limb, being connected to it continuously' [3] (p. 22). Such developments invite a much-needed critique, not only as to how technology mediates our sense of self and what it means to be human, but also how it might be used to redesign the human form and its existence.

Such questions do not belong in futuristic fiction: virtual realities, genetic engineering, anti-aging treatments and performance-enhancing drugs, bionic prosthetics, and citizenship rights for Sophia the cyborg are already redefining humanity and what it means to be human. A friend of mine has implanted a microchip the size of a grain of rice under his skin, so that his 'hand' is able to communicate directly with my smart phone—it makes for a fun party trick. Reporting to Euronews, an IT planner described it as '...a step towards the future... It is extremely futuristic although it is already happening. This technology was born to help us, to give us small 'superpowers" [40]. Such developments can be seen as part of the transhumanist agenda that seeks to extend the relationship between the human and technology, merging the two, thus making enhancement options available to all persons as we navigate the transition to the posthuman as they imagine it [36]. Snaza and Weaver [41] argue that young people today are already posthuman, the 'merger of human with machines that extends, supplements, and enhances human capabilities, including extended life expectancies, enhanced learning capabilities, supplemented body forms and conditions, and extended capabilities for control of bodies and minds' (p. 350). They make the point that this program of human enhancement, toward cyborg, is only available to those who can pay for it, whilst also coming at a non-financial cost. Humanism, they argue, driven by the meta-narrative of Homo economicus, has abandoned the human.

So what is the value of the 'human' in such a world? What if the digital divide produces a hierarchy of 'being human'? Or does this exist already? Are the rights we are extending to Sophia the cyborg denied to some human beings? Young [42] describes youth work as 'an exercise in moral philosophy' (p. 3). In this paper I hope to suggest a philosophical and theoretical orientation that might provide youth workers with a lens to frame such ethically loaded questions; but first, allow me to introduce another Sophia, the illegal baby.

## 4. A Tale of Two Sophias: Sophia the Illegal Migrant Baby

Sophia was the name given to a baby girl born on a German military warship that bears the name of a Prussian princess, her namesake, Sophia. Baby Sophia was born on 24 August 2015, to a Somali mother. EUNAVFOR MED Operation Sophia is a Common Security and Defence Policy operation tasked with disrupting smugglers' and traffickers' operations, and part of EU efforts to restore security in Libya and the Central Mediterranean [43]. The operation itself is named after baby Sophia. But why was Sophia born on a military frigate in the middle of the Mediterranean?

As the scale of international migration has continued to grow, the number of people who have been forced to flee their home has also grown. The nature of the threat people face has also transformed, and so people continue to flee their country and cross borders as a result of war, conflict, economic and political instability, environmental change and the broader impacts of climate change, food insecurity, and the perceived and very real disparities between the global haves and have nots. Global capitalism and the ongoing demand for cheap labor and exploitation of natural resources and land continues to determine the lives of young people around the world. The financial crisis of 2008 resulted in massive youth unemployment around the globe, and many of these young people resorted to migration in search of employment opportunities beyond their national borders. The IOM World Migration Report [44] positions contemporary migration patterns within a 'period of considerable uncertainty' marked by a questioning of the global political order that was forged over the last century and growing frustration with the failings of liberal rationalism (p. 1).

Indeed, a growing evidence base (and, indeed, media headlines around the world) demonstrates how migration is linked to broader global economic, social, political, and technological transformations that influence a spectrum of prioritized policy issues and our daily lived experiences. These same transformations have been identified as a source of national anxiety, particularly for the richer regions of the world, including the European Union and North America. Migrants, particularly those from the global South, the colonial script still etched deep into their dark bodies, have been positioned as convenient scapegoats by governments and far-right parties (which are sometimes both) as they draw on nationalistic and ethnocentric rhetoric calling to 'take back control' of borders, sovereignty, and nation.

For some young people, particularly those from the global South, opportunities for legal travel have become increasingly restricted, resulting in increasingly violent border practices. IOM's 2020 report on Child Migration draws attention to 'the absence of adequate legal pathways for the exercise of child and youth mobility' and how this contributes to dangerous and often deadly migration and the 'illegalization' of the migrant (see also Pisani [27]).

Bauder uses the term 'illegalized' to draw attention to processes, specifically the 'institutional and political practices', that render people 'illegal' and thus puts them in more danger (p. 327).The increasingly complex political economy that transcends national borders, expanding to every corner of this planet, has produced a cold brutality. At the time of writing, since 2014, more than 25,000 refugees and migrants had lost their lives trying to cross the Mediterranean Sea by boat [45]. Over the years, the journey has become even more dangerous as a direct result of EU policy, including cutting back on state-led search-and-rescue missions, ongoing efforts on the part of Member States to thwart NGO search-and-rescue operations, and increasingly dangerous smuggling operations including the use of unsafe boats. Constructed as a 'threat' by politicians eager to win their next term, the life of the 'illegal immigrant' is rendered both exploitable and disposable. The chilling words of Hannah Arendt [46] resonate; recounting the Nazi atrocities of WWII, she concluded that the 'world found nothing sacred in the abstract nakedness of being human' (p. 299). Whether it be the US–Mexico border, the Sahara Desert, the exodus of the Rohingya, or the Mediterranean Sea, thousands continue to die in their attempt to reach

security and the hope of a better life, and thousands continue to be denied the right to rights, as they are, it would seem, not human enough.

So, we are not all in the same proverbial boat. As some of us may have been contemplating which part of this beautiful planet to explore next, baby Sophia was born on a frigate because her mother, a Somali national who undoubtedly would have been in search of safety and a better life, would have been denied the possibility of traveling in a safe and legal manner, and thus, illegalized.

## 5. Critical Posthuman Youth Work: Reimagining Relational Praxis

The posthuman convergence offers opportunities for new ways of thinking about young people and youth work. Rather than taking the route of apocalyptic panic, this predicament can provide opportunities to think through how youth work might respond to the challenges and contradictions that define this era. In the final section of this paper, I propose critical posthuman theory as an epistemological lens to reconsider the values that underpin our work, and new ways of thinking about youth work. Youth work, through a critical posthuman lens, remains grounded in the day to day, embodied and embedded experiences of young people, and starts where young people are at, but with a new critical awareness of, and respect towards, our tangled interdependence with the human and non-human (including technological) others.

Whilst there can undoubtably be some strategic, political benefits to working with a fixed 'category' of 'youth' to address ongoing structural injustices and violence, a critical posthuman conceptualization of youth moves beyond fixed, age-specific or sovereign identities. From a critical posthuman perspective, there is no essence to being a young person, and young people are not impervious to transformation. Such conceptualization provides the possibilities for alternative representations and lines of flight that may emerge as creative forces beyond the binaries of child/adult, nature/culture, or object/subject.

Young people are understood as material, relational agentic and mediated beings. Already human and machine, bodies merged with technology and enmeshed within economic relations that are driven by the logic of advanced capitalism and the maximization of profit. As posthuman subjects, all young people are engaged, symbiotically and differentially, in multiple and intertwined processes of becoming, with the ability to affect and be affected. Neither exceptional nor autonomous, they are embedded in a network of complex and multifaceted relationships, becoming within the evolving historical context, with their environment, with non-human life, with—and of—the planet.

As non-unitary cultural and social agents, young people—to different degrees and intensities—experience, shape, and are shaped by historical processes, depending on the degree to which they are valued, or not. As a social construction, the meaning and experience of youth is also understood to be interdependent, materially embedded and embodied within, and subject to, multiple ongoing and evolving institutional figurations—including youth work—and the intersections of dominant narratives, including neoliberalism, racism, agism, homophobia and sexism, among others. As documented in this paper, the ongoing reports of violence experienced by so many of the human species demonstrate how many young people have been excluded from the category of 'Human' altogether [27]. As such, a critical posthuman lens might look to an understanding of young people that seeks to favor processes of becoming—within these differential politics of location—over identities as essence, that all too often serve to reinforce and perpetuate hierarchies grounded in such violent structural forces. What I am describing here is an understanding of young people as subjects in movement, subjects in the process of becoming (Braidotti, 2019) [14]. This understanding of youth, as simultaneously being transformed and transforming, affected and affecting, provides alternative possibilities for subjectivity, ethical frameworks, and hopes and desires.

From a critical posthuman perspective, youth work is also embedded in these relationships, enmeshed in processes with other institutions and the historical, political, economic, social, ecological, and technological context. It is this reality, this understanding, that offers

youth work new possibilities and opportunities to think through the posthuman predicament and find other ways that look to life and love, rather than violence and destruction. For example, within the youth work process, we might explore how these relationships are experienced differently, and why. As an ethical praxis, then, youth work's commitment to 'tipping' the balances of power in favor of young people [47] (p. 103) must also reflect on broader power relations and affects. Thus, such an approach begs the question, 'which young people would benefit from tipping the balance of power?' The young person, as a knowing subject, is encouraged to engage in a pedagogical process that seeks to step away from the normative Eurocentric humanist and anthropocentric lens, to decenter the self, and critically reflect on one's position (and privileges) within broader power relations.

Moving beyond the notion that youth work should remain an expression of the 'centuries-old, humanist project' [48] (p. 125), a critical posthuman approach to youth work might also challenge the dominant paradigm that youth work remains 'primarily concerned with people' [5] (p. 2). A critical posthuman approach to youth work requires a radical repositioning of youth, a decentering that simultaneously also expands ethical accountability beyond the human and individualist autonomy, creating assemblages of human and non-human others. It also provides new possibilities to get creative and experiment with issues such as ecological justice, advanced capitalism, social in/justice, equity and institutional encounters, environmental wellbeing, the morphing of the human and technology, and what such bodies are capable of being and of becoming. Critical posthuman youth work, then, has the potential to forge new alliances with vibrant assemblages that seek to respond to the local environment and beyond, in which young people are entwined within the youth work processes, together transforming and being transformed.

Relationships are familiar territory for youth workers [49], but the posthuman predicament alerts us to transcend the traditional focus of belonging, to reach out beyond the local, transgressing binaries and boundaries—be they human, geographical, political, virtual/technological, ecological, animal, or other. It invites us to consider our ethical responsibility towards someone who may live on the other side of the planet, someone who I will never know, and yet with whom I am intimately connected. For example, if we were to think about the young person, working on the factory line, manufacturing the garment I ordered online, how are we to understand and make sense of this complex and multifaceted relationship between us both? How do economics, technology, global youth culture, environmental concerns, and historical relationships weave through this relationship? Likewise, and as already argued by Skott-Myhre and Skott-Myhre [20], the relational nature of youth work can be more inclusive in a way that recognizes the 'entanglements of interspecies relations with the economic imperatives' of a given space, within its historical context. Within a youth work context, we might ask, for example, how do my 'needs' impact the local flora and fauna? How does their wellbeing affect me? This points towards an expansion of the community of practice, if you will.

If youth work is about accompanying young people as they make sense of the world, and their evolving place in that world [42], then youth work that adopts a critical posthuman lens is offered here as an opportunity for youth workers, and young people, to think through, and perhaps act on, the posthuman predicament. This is an approach that looks to multiple relationships and processes, and how we are becoming together, through, and within these interactions.

It may provide the potential to reflect on how young people relate with each other within our technologically mediated realities—in many ways, becoming cyborgs—as individuals, as communities, as citizens—or not—of nations, and as a species. Youth work, in this way, can be understood as an ethical process that seeks to understand existing and evolving power relationships, including our differential relationship with capitalism, the ongoing exploitation of human and non-human resources, and the violence produced within and by this system, that demarcates the racialized 'other' as the less-than-human, including baby Sophia.

If, as suggested by Seal [5], 'youth and community workers operate through words and ideas' (p. 2), a posthuman critical approach can work towards generating new ways of imagining the human subject in the world. The predicament in which we find ourselves offers youth workers with opportunities to take different lines of flight, to push the boundaries of theory in ways that might respond to the complexities and contradictions that weave through the posthuman predicament. It offers opportunities to consider alternative ethical paradigms that might frame youth work processes in ways that are productive and creative, transforming relationships and spaces. A critical posthuman lens offers the possibilities to construct and nurture new relational encounters that are open to new possibilities. In thinking of the present as the virtual future, youth work can engage in a process of becoming, and this in itself is praxis. Change is located in our actions today and every day, in our relationships with the human and non-human other, near and far.

Our journeys are inter-woven, and it is this discernment and understanding of our mutual bind that may also be the source of our liberation: a pedagogy of hope, a praxis that looks towards alternative ways of becoming as both unique, complex and evolving subjects, and technologically mediated and interdependent beings. In adopting a critical posthuman lens, youth work is re-affirmed as a political practice that embraces the messy borderlands, the spaces in-between [12,29], and builds on the work of critical pedagogies (Freire and bell hooks immediately come to mind). Such an approach offers an inclusive democratic and critical process that is open to new possibilities, and new ways of being in the world that requires continuous ethical negotiation and reflection, and multiple forms of accountability. This is a creative, affirmative process. It is also a radical process, one that is grounded in a love that transcends family, community, nation, and species. It is a love grounded in difference.

## 6. Conclusions

Our journeys into the posthuman predicament are diverse, complex and intersectional, embodied, and historically and geopolitically situated. The effects of climate change and the sixth extinction, advanced capitalism, globalization and the deep divides that it carves, the 4IR, automation, digitalization, and developments in AI, on both the human and non-human, are experienced differentially according to the historical, social, economic, legal, and geopolitical markers and borders that continue to distinguish and discriminate. We are all enmeshed and entangled: young people in the local youth club, the lungs of the earth, technological agents including algorithms and humanoid robots such as Sophia, and Sophia the illegalized baby, born on a military vessel that was positioned in the Mediterranean to protect 'us' from 'them'.

Embracing 'sophia' as wisdom, embodied as both knowledge and experience in a creative process, provides the possibility of regarding the posthuman condition as an opportunity to raise new questions and explore alternative ways of becoming.

Can youth work provide the space and opportunities for young people to think 'in and with the world' [14] (p. 143)? For thousands of years, philosophers have grappled with what it means to be human. Youth work, as an 'exercise in moral philosophy' [42] (p. 2), provides the space, opportunity, and skills for young people to question. I believe youth work is well positioned for, and has a vital role in, creating spaces and opportunities for young people to grapple with the different futures that lie ahead of them, with the ethical issues that will arise and decisions that will need to be taken, before someone else (and here I include Artificial Intelligence) takes the decisions on their behalf.

Youth work as a critical posthuman political project can continue to create those spaces and opportunities to engage in dialogue, think critically, and act in order to transform. If youth work is to remain relevant and responsive to young people's lives, it must respond to embodied and embedded, individuated and situated realities. It needs to be a relational practice located in the local, and also the global, transcending borders and positioned within our shared (albeit not equally) planetary condition.

In this paper, in an effort to open debate, I have reflected on, and offered a consideration of how, youth work, framed within a critical posthuman paradigm and adopting new materialist methodologies, might consider, question, engage with, and respond to the posthuman convergence. This includes the complex and evolving social, political, environmental, technological, and subjective changes that are both situated and inter-related, local, transnational, and planetary. I position a critical posthuman approach to youth work as a creative process, one that embraces imagination and new possibilities. Hope is a political act.

**Funding:** This research received no external funding.

**Institutional Review Board Statement:** Not applicable.

**Informed Consent Statement:** Not applicable.

**Data Availability Statement:** Not applicable.

**Conflicts of Interest:** The author declares no conflict of interest.

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
