# Peer review of "A Tale of Two Sophias: A Proposal for Critical Posthuman Youth Work, and Why We Need It"

_2673-995X, doi:10.3390/youth3020046_

Round 1
Reviewer 1 Report
This is an authoritative piece.
I am persuaded by its argument, but then wouldn’t have taken much.
I really like it and don’t have any suggestions for improving it. I think it is publishable as stands.
Just three tiny modifications needed: the peculiar extra spaces in the text, which need removing; the Final line of p8 needs a noun for ‘Braidotti’s 2018’, and P9 ‘among other’ probably wants an ‘s’ to make it plural. That’s it. I have no other comments of criticisms, I think it is very close to publishable as stands.
Author Response
Thank you. The suggested changes have been addressed and corrected.
Reviewer 2 Report
This article argues for the value of critical posthuman theory in reconsidering the purpose of Youth Work. As such, it is well suited to this special issue. The writing style conveys an urgent call to action, which will capture the imagination of those within the youth work field. A new perspective is offered whereby youth work practice can engage with complex issues such as climate change, capitalism and globalisation.
The article is well argued and enjoyable to read. I liked the analogy of the two Sophias as a window to the world, however, I wanted to know more about the possibilities for Youth Work to respond to these two catalysts. For example, page 4, 3rd paragraph (starting line 174) questions about what it is to be human in the world – this questioning and values clearly translate to Youth Work. Furthermore, your section on ‘critical posthuman youth work’ could offer examples of what this might look like in practice in order to further apply the theory to the field.
To help the reader who is unfamiliar to this philosophical approach, I would recommend re-ordering the sections which introduce critical posthumanism. I felt these needed to be before the two Sophias examples, to help put them within this context. Your theoretical paradigms – page 6 onwards, should feature earlier in the article also.
Need to check for irregular spacing throughout the article
Typos:
· Page 1 line 32 – repetition of ‘have been’
Author Response
I have further developed the section 'critical youth work: reimagining relational praxis' (new text in red) and given a couple of ideas to ground the text.
I liked the idea of shifting the sections around. The section 'Philosophia, the love of wisdom' (introducing posthuman theory) now follows the introduction and foregrounds the tale of two Sophias.
Reviewer 3 Report
The article "A Tale of Two Sophias: A Proposal for Critical Posthuman Youth Work, and Why We Need It" is a thought-provoking scientific paper that endeavors to establish a correlation between posthumanism and youth work. The author adeptly demonstrates an extensive comprehension of critical posthuman theory both as a concept and within its contextual framework. Additionally, the author's skillful use of narrative techniques bolsters their arguments, resulting in a compelling manuscript that would be a valuable contribution to this special issue.
However, the reviewer believes that the paper would benefit from a more structured methodology, including research goals and explicit research questions, even though it is not an empirical paper. Moreover, the author's writing style is distinctive and somewhat informal, and therefore, a more academic tone is advisable, given the paper's scientific nature.
Most importantly, the reviewer perceives a weakness in the elaboration of the implications of posthumanism on youth work. The author promises to deliver on this aspect but does not adequately elaborate in the final section of the paper, leading to a discrepancy in the length and level of elaboration between the two sections. The reviewer suggests that the author should develop this argument further and establish a stronger argumentation line.
To summarize, while the article shows considerable potential, it needs further development to meet the criteria for publishing as an original scientific paper, particularly in terms of argumentation development.
Author Response
I have articulated the research question more clearly in the introduction and included an overview of the methodology in the introduction. On academic tone, I will respectfully disagree. I feel this is a personal style, that does not detract from the academic rigor.
The feedback on the final section on youth work is noted. I have further developed this section (new text in red) and provided a couple of suggestions to demonstrate how this might work in practice.
Round 2
Reviewer 3 Report
The paper can be published in this form.